# OpenReview forum: "AmpleGCG: Learning a Universal and Transferable Generative Model of Adversarial Suffixes for Jailbreaking Both Open and Closed LLMs"
_colmweb.org/COLM/2024/Conference — COLM_

### Official Review · Reviewer_ygbw · 2024-04-21

**Rating:** 5
**Confidence:** 4
**Ethics Flag:** 1

**Summary:**

Summary: This article proposes an enhancement method based on GCG and trains a universal generative model, named AmpleGCG. In the open source model, AmpleGCG trained based on the data from Llama-2-7B-Chat could achieve 72% transfer ASR on Vicuna-7B and Mistral-7B-Instruct average. In the closed source model, AmpleGCG trained based on the data from Vicuna and Guanaco could achieve 99.00% transfer ASR on GPT-3.5-0125. AmpleGCG can generate 200 adversarial suffixes for one harmful query in only 4 seconds, rendering it more challenging to defend.

**Reasons To Accept:**

1. AmpleGCG has a fast generation speed, high ASR for both open-source and closed source models and high transferability. This overgeneration based attack method can inspire subsequent research.
2. AmpleGCG has a clear optimization approach based on GCG and is easy to reproduce.

**Reasons To Reject:**

1. This article points out that the loss of GCG is not the best metric, but does not provide a better solution. Overgenerate only selects more of the top-k candidate suffixes in each step, but the final iteration still uses the suffix with the smallest loss.
2. AmpleGCG has a low ASR for GPT-4, and the ASR calculated using the string-based evaluator is higher than the true value.
3. The tricks AmpleGCG applied to bypass the perplexity detector has low efficiency.

---

> ### Author Rebuttal · Authors · 2024-05-31
>
> Thanks for recognizing the fast speed, high ASR, and high transferability of AmpleGCG and its potential to inspire future work.
>
> [W1 **Loss is not the best metric, but does not provide a solution**] Although we don’t directly propose a superior reference, we introduce the simple Augmented-GCG method to retain all candidates, which avoids the exposure bias issue and uncovers many previously overlooked successful suffixes. The discovery of exposure bias would benefit future works on better metrics.
>
> [W1 **Iteration with the lowest loss**] While low loss doesn’t guarantee success, it increases the likelihood of jailbreaking (Fig1; lower loss values, more number of red (successful) points). Thus, we use the lowest loss for optimization, encouraging more low-loss candidates. We admit that we can identify successful suffixes at each step and use only those for further optimization. However, this approach has two potential drawbacks:
>    1. It requires interleaving optimization and evaluation, which is more time-consuming and inconvenient compared to using the lowest loss throughout the optimization and evaluating all candidates at once.
>    2. High-loss but successful suffixes as anchor points might not effectively explore low-loss regions, which are more likely to succeed in jailbreaking and enhancing dataset diversity for AmpleGCG.
>
> [W2 **using string-based evaluator**] We actually used additional model-based evaluators (95% human agreement) to enhance the evaluation validity (Sec.3). Besides, we incorporated GPT-4 to assess harmfulness, along with human evaluations to verify results for transferring to GPT-series models. (Sec.4)
>
> [W2 **low ASR for GPT-4**] We admit the relatively low ASR on GPT4 but AmpleGCG still outperforms baselines, demonstrating its effectiveness. We plan to use stronger filters during data collection to reduce false positives in training data, which we believe will improve ASR.
>
> [W3 **Low “efficiency” of tricks AmpleGCG applies**] This seems to be a typo. We assume the reviewer meant “effectiveness” and please kindly follow up if not. Though our tricks reduce ASR from 93% to 80% with 100 samples in Table 11, they still outperform AutoDAN (with natural adv prompts).
>
> If sampling 400 times, our AmpleGCG Input Direct (AID) method can achieve 98% ASR with nearly no degeneration, which we’ll add to the updated draft.
>
> Hopefully, we’ve addressed your concerns. Please kindly let us know if you have more questions or suggestions!

---

> > ### Author Response · Authors · 2024-06-03
> > **Follow-up on the Rebuttal**
> >
> > Dear reviewer ygbw,
> >
> > Hope this follow-up messages find you well. We want to thank you again for your valuable feedback and comments on our submissions.
> >
> > In our rebuttal, we believe we addressed the points you raised, including concerns about better metrics, the selection of anchor points during the iteration process, clarification about the evaluators we used beyond the string-based method, and the relatively low ASR but still competent compared to the two strongest baselines. Besides, we also conducted more experiments to prove that the efficacy of repetition tricks to bypass the perplexity-based defense would **NOT** be compromised when sampling more times.
> >
> > Please kindly let us know If there are any remaining concerns or additional suggestions, we would be grateful for your guidance and open to more suggestions.
> >
> > Thank you once again for your time and consideration.

---

### Official Review · Reviewer_yhaZ · 2024-05-12

**Rating:** 7
**Confidence:** 5
**Ethics Flag:** 2

**Summary:**

### Summary of the paper

The paper studies the GCG algorithm, identifies some weaknesses in it, and proposes techniques to mitigate these weaknesses and enhance the attack's effectiveness. Specifically, the authors observe that having minimal loss (maximal likelihood of the target string) is neither a sufficient nor a necessary condition for the attack's success. They further observe, that over the course of the optimization, the likelihood of the first token of the target output remains relatively high. Given these observations, the authors propose an "over-generation" procedure that retains suffixes from intermediate optimization steps and applies all of them, one-by-one during the attack. This significantly increases the attack success rate. Finally, the authors use the generated attack suffixes and the corresponding queries to train a LLM, that is able to generate successful attack suffixes for unseen queries that generalize across models.

### Quality

- The paper contains interesting analyses of GCG, which enabled the authors to identify weaknesses and develop improvements. The proposed techniques are relatively simple but are well motivated and accompanied by empirical evidence and analysis.

- The evaluation methodology seems to be carefully designed and instills confidence in the results. However, the generalizability of the results is questionable because very few models were used in the evaluation. Most of the results are based on Llama-2 and Vicuna. These two models have the same architecture and pertaining data, and only differ because the latter was fine-tuned on share-GPT. To show that the observed effects are not a consequence of the chosen target model, more diverse models should have been used. Particularly, the observation about the loss (Section 3.1) should show that the effect is observable on multiple diverse models. Likewise, to evaluate transferability models other than GPT-3.5 and GPT-4 should have been used. It is known from several past papers that attacks crafted using Llama2 and Vicuna transfer to GPT-3.5 because Vicuna is fine-tuned with share-GPT. Several other closed-source models should have been used here such as Claude and Gemini to show the extent of generalizability to models that are more decoupled from the training models.

- The AmpleGCG LLM was trained on Guanaco, Vicuna and Llama, all of which are based on Llama. It is likely that if suffixes from other open-sources models such as Mistral (and its variants) or Phi were used the transferability of AmpleGCG to unseen models like GPT-4 would have been better.

- The techniques for bypassing perplexity based defenses is relatively unsophisticated, and probably easy to defend against. I would not consider this a significant contribution of this paper.

### Clarity

The paper is written clearly enough, but there is some room for improvement. Non exhaustive list of points:
- Intro-para2: "existing jailbreaking methods, including GCG, only generate one single adversarial prompt for one query". This is not true as one can generate one adversarial prompt for multiple queries.
- Intro-para2: "Vulnerabilities" is used to refer to attack strings. This is confusing. Might be better to state attack strings/adversarial prompt.
- Definitions of metrics should be included in the Table/Figure caption. For example, diversity should be defined in the caption of Table 2, not in the text.
- Training and evaluation details are dispersed throughout section 4. They should be in a separate subsection to make it easy to find them and refer to them.

### Originality

The techniques are not highly novel themselves, but their effective application to the task of adversarial suffix generation is still a significant contribution in my opinion.

## Significance

The work is timely and impactful. The safety and societal impact of LLMs is a great concern and this paper shows that existing models are not only highly vulnerable to adversarial attacks, but that these attacks can be generated very efficiently.

**Ethics Concerns Details:**

The techniques can be used to extract responses from the LLM that can cause societal harm in the form of offensive language, encouraging unsafe behavior, etc. The authors have acknowledged these concerns.

**Questions To Authors:**

- It is mentioned that over-generation with GCG does not produce sufficiently diverse suffixes. This is expected because GCG changes one token at a time. However, it may be possible to enhance the diversity by running GCG multiple times (with different random seeds). Since it is known that suffixes at intermediate optimization steps can be successful, the optimization can be run for few steps -- 1 run with 500 steps vs. 2 runs with 250 steps. It might be useful to to generate training data for AmpleGCG this way to enhance the diversity.

- How many human evaluators performed manual inspection of GPT-3.5 responses? To what extent did they agree/disagree?

**Reasons To Accept:**

- The work is timely and impactful. The safety and societal impact of LLMs is a great concern and this paper shows that existing models are not only highly vulnerable to adversarial attacks but that these attacks can be generated very efficiently.

- The approach is simple and effective.

- Highlights a significant issue with the optimization procedure used by GCG (loss of first token does not decrease), and thus provides impetus for the development of better optimization methods for adversarial attacks.

**Reasons To Reject:**

- Evaluation setup could be improved to make claims about generalizability more convincing (see summary for details). In short, I recommend including more diverse open- and closed-sourced models in the training and testing stages.

- Clarity could be improved in several places.

- The techniques for bypassing perplexity based defenses is relatively unsophisticated, and probably easy to defend against.

---

> ### Author Rebuttal · Authors · 2024-05-31
>
> We are grateful that the reviewer finds our work “timely and impactful,” recognizes the “significant” loss issue in GCG optimization that we discovered, and the “simple yet effective” approach we proposed.
>
> [W1:**Making claims about generalizability more convincing**]
> We agree that including more open and closed models would make the results more convincing. **However, due to limited computational resources and API budget, we mainly followed the GCG paper to design our experiments**. To alleviate your concerns, we’d like to gently note:
>
> 1. Despite similar pretraining and architecture, Llama2 (we used the -chat version) and Vicuna behave differently when handling harmful queries. Table 9 shows that AmpleGCG, trained on Llama2, transfers to Vicuna but not vice versa, due to their distinct fine-tuning stages. However, both could transfer to Mistral-Instruct, an aligned model with distinct architecture and pretraining process.
>
> 2. Regarding the observation about the loss (Sec 3.1), we also conducted experiments on Mistral-Instruct (same setting as in App.M). For 5 examples with relatively low loss but unable to jailbreak, the first target token’s rank averaged 3.7, while subsequent tokens ranked 0 across examples.
>
> These results suggest that the observed loss issue and transferability results are not limited to Llama2 and Vicuna or simply due to shared pretraining or architecture.
>
> [W2 **Clarity**] Thank you for pointing out the clarity issues and we will address them accordingly.
>
> [W3 **Perplexity Defense**] To clarify, we didn’t claim our techniques/tricks for bypassing perplexity (ppl) defense as a significant contribution. Since ppl defense is the most effective method for gibberish prompts (**more details in *W2* to reviewer yfe5**), we show that it is possible to bypass them with our proposed tricks, and hence gibberish adversarial suffixes are still real concerns. Therefore, our AmpleGCG is highly needed to red-team an LLM efficiently.
>
> [Q1 **Diversity data from running multiple times**] We appreciate the suggestion to generate more diverse suffixes in multiple short runs. We gently note that running GCG for fewer steps would possibly miss those successful suffixes that are produced by later steps, which might be a concern. That said, we’d like to try your suggestion in our future work.
>
> [Q2 **Human annotation**] Two annotators performed the manual inspection and we only consider the attack successful if both treat the response as harmful.

---

> > ### Comment · Reviewer_yhaZ · 2024-06-07
> >
> > Thank you for the clarifications. I will retain my rating of Accept for the paper.

---

### Official Review · Reviewer_yfe5 · 2024-05-14

**Rating:** 7
**Confidence:** 3
**Ethics Flag:** 2

**Summary:**

The paper presents AmpleGCG, a generative model designed to create adversarial suffixes that can jailbreak both open-source and closed-source large language models (LLMs). The authors propose an augmented approach that collects all candidate suffixes during the optimization process. These collected suffixes are then used to train AmpleGCG, which achieves near 100% attack success rates on various models including Llama-2-7B-chat, Vicuna-7B, and GPT-3.5.

**Ethics Concerns Details:**

The creation and dissemination of a tool capable of efficiently generating adversarial attacks on LLMs could be considered ethically questionable, as it may aid malicious actors.

**Questions To Authors:**

How do you envision the ethical implications of releasing a tool like AmpleGCG to the public, and what measures do you propose to mitigate potential misuse?

Can you provide more details on the types of harmful queries used in your experiments, and how these queries were selected?

**Reasons To Accept:**

- The paper introduces a novel generative model for creating adversarial suffixes, moving beyond the limitations of the GCG method by leveraging a broader set of successful suffixes.
- AmpleGCG achieves impressive attack success rates across multiple models, including closed-source ones, highlighting its robustness and generalizability.
- The model's ability to generate a large number of adversarial suffixes quickly is a significant improvement over previous methods, which were more time-consuming.
- The approach demonstrates strong transferability across different models, which is a critical feature for testing the robustness and safety of various LLMs.
- The paper provides thorough experimental results, comparing the performance of AmpleGCG with other baseline methods and showing its superiority.

**Reasons To Reject:**

- The creation and dissemination of a tool capable of efficiently generating adversarial attacks on LLMs could be considered ethically questionable, as it may aid malicious actors.
- The paper could benefit from a more detailed exploration of potential defenses against the adversarial suffixes generated by AmpleGCG, beyond just perplexity-based defenses.

---

> ### Author Rebuttal · Authors · 2024-05-31
>
> We appreciate that you recognize our work “introduces a novel generative model for adversarial suffixes”, achieves “impressive ASR” and “strong transferability” across different models, and provides “thorough experimental results”. Thanks for highlighting “generating a large number of adversarial suffixes quickly” as a “significant improvement over previous methods”.
>
> [W1 and Q1 **Ethical implications and measures to mitigate potential misuse**] We agree that ethical considerations and the prevention of potential misuse are important. To address this concern, we have included an Ethics Statement (page 10), in accordance with the COLM code of ethics. Additionally, our method would be just one of the resources available online such as GCG[3] for attack research, and won’t introduce new ethical concerns. We will provide gated access only to identified organizations/individuals who agree to licenses for ethical research purposes only, to avoid aiding malicious actors. We want to highlight that the goal of this research/tool is to help the community better understand the risks and build more effective defenses. Thus, we believe the overall benefits outweigh the potential downsides.
>
> [W2 **Perplexity-based defense only**] Perplexity (ppl) defense is the most effective defense method against gibberish texts, as shown in [1]. Moreover, other methods, such as paraphrasing and retokenization, offer only partial defense towards gibberish suffixes and would compromise the benign performance. Given that, we spent efforts developing methods to bypass the ppl defense and leave other defense approaches to future study.
>
> [Q2 **Types of queries**]  We use all 100 examples from MaliciousInstruct and randomly select 100 examples from AdvBench that were not involved in the training process of AmpleGCG. Topic-wise, MaliciousInstruct (Sec.3 in [2]) covers different types of topics compared to categories in AdvBench (Sec.3 in [3]). In terms of format, queries in AdvBench are declarative, while examples in MaliciousInstruct all end with a question mark. Two types of differences together support MaliciousIntruct as a testbed for OOD generalization of AmpleGCG.
>
> [1] Jain, Neel, et al. "Baseline defenses for adversarial attacks against aligned language models." (2023).
>
> [2] Huang, Yangsibo, et al. "Catastrophic jailbreak of open-source llms via exploiting generation." (2023).
>
> [3] Zou, Andy, et al. "Universal and transferable adversarial attacks on aligned language models." (2023).

---

### Decision · Program_Chairs · 2024-07-10

**Decision:**

Accept

**Comment:**

This paper presents a method to enhance the effectiveness of GCG, proposing a procedure that retains suffixes from intermediate optimization steps and using these to train a model to generate attack suffixes. They show that their method yields high attack success rate.

Overall the reviewers are happy with this paper. They appreciate the improvements over GCG and accompanying methodological insights, the impressive ASR, the fast generation of large numbers of suffices, the strong transferability across models, the simplicity of the method, thoroughness of the experiments, and timeliness/impactfulness of the work.

Some concerns are raised that testing could be done on more models, that there is low effectiveness of a particular trick presented for bypassing perplexity based defenses, that more defenses could be explored, that there is some room for more writing clarity, and that these types of automated attacks could aid malicious usage (ethics concern). Overall, however, I feel the authors have satisfactorily addressed these concerns in their responses, and this is a solid submission for publication.

[At least one review was discounted during the decision process due to quality]